# Automatic Curriculum Learning through Value Disagreement

**Yunzhi Zhang**
UC Berkeley

**Pieter Abbeel**
UC Berkeley

**Lerrel Pinto**
UC Berkeley, NYU

## Abstract

Continually solving new, unsolved tasks is the key to learning diverse behaviors. Through reinforcement learning (RL), we have made massive strides towards solving tasks that have a single goal. However, in the multi-task domain, where an agent needs to reach multiple goals, the choice of training goals can largely affect sample efficiency. When biological agents learn, there is often an organized and meaningful order to which learning happens. Inspired by this, we propose setting up an automatic curriculum for goals that the agent needs to solve. Our key insight is that if we can sample goals at the frontier of the set of goals that an agent is able to reach, it will provide a significantly stronger learning signal compared to randomly sampled goals. To operationalize this idea, we introduce a goal proposal module that prioritizes goals that maximize the epistemic uncertainty of the Q-function of the policy. This simple technique samples goals that are neither too hard nor too easy for the agent to solve, hence enabling continual improvement. We evaluate our method across 13 multi-goal robotic tasks and 5 navigation tasks, and demonstrate performance gains over current state-of-the-art methods.

## 1   Introduction

Model-free reinforcement learning (RL) has achieved remarkable success in games like Go [49], and control tasks such as flying [26] and dexterous manipulation [4]. However, a key limitation to these methods is their sample complexity. They often require millions of samples to learn a single locomotion skill, and sometimes even billions of samples to learn a more complex skill [7]. Creating general purpose RL agents will necessitate acquiring multiple such skills, which further exacerbates the sample inefficiency of these algorithms. Humans, on the other hand, are not only able to learn a multitude of different skills, but are able to do so from orders of magnitude fewer samples [25]. So, how do we endow RL agents with this ability to learn efficiently?

When human (or biological agents) learn, they do not simply learn from random data or on uniformly sampled tasks. There is an organized and meaningful order in which the learning is performed. For instance, when human infants learn to grasp, they follow a strict curriculum of distinct grasping strategies: *palmar-grasp*, *power-grasp*, and *fine-grasp* [33]. Following this order of tasks from simple ones to gradually more difficult ones is crucial in acquiring complex skills [38]. This ordered structure is also crucial to motor learning in animals [50, 28]. In the context of machine learning, a learning framework that orders data or tasks in a meaningful way is termed 'curriculum learning' [12].

Most research into curriculum learning has focused on the order of data that is presented to a supervised learning algorithm [15]. The key idea is that while training a supervised model, 'easy' data should be presented first, followed by more difficult data. This gradual presentation of data is shown to improve convergence and predictive performance [12]. However, in the context of reinforcement learning, how should one present a curriculum of data? The answer depends on what aspect of complexity needs to addressed. In this work, we focus on the complexity involved in solving new tasks/goals. Concretely, we operate in the sparse-reward goal-conditioned RL setting [45]. Here, the

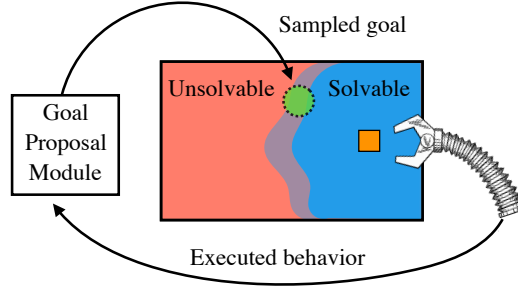

Figure 1: In this work we focus on generating automatic curriculums, where we propose goals that are right at the frontier of the learning process of an agent. Given trajectories of behavior from a goal-conditioned RL policy, our value disagreement based Goal Proposal Module proposes challenging yet solvable goals for that policy.

sparse-reward setting reflects the inherent difficulty of real-world problems where a positive reward is only given when the goal is achieved.

To improve the sample efficiency of goal-conditioned RL, a natural framework for using curriculums is to organize the presentation of goals for the RL algorithm. This goal proposer will need to select goals that are informative for policy learning. One option for the goal proposer is to sample goals that have been previously reached [3]. Hence, as the algorithm improves, the sampled goals become more diverse. However, this technique will also re-sample goals that are too easy to give a useful training signal. The central question to improving the goal sampler is hence, how do we select the most useful and informative goals for the learning process?

To sample relevant goals that are maximally informative for the learning process, recent work [51, 39] focuses on using adversaries to sample goals for the agent at hand. Here, the adversary samples goals that are just at the horizon of solvability. These goals form a powerful curriculum since they are neither too easy nor too hard and hence provide a strong learning signal. However, due to the instability in adversarial learning and extra samples needed with multiple agents, these algorithms do not scale well to harder problems. Moreover, setting up an explicit two-player game for different problem settings is not a scalable option.

In this work, we propose a simple, but powerful technique to propose goals that are right at the cusp of solvability (see Figure 1). Our key insight is to look a little closer at the value function. In goal-conditioned settings, the value function of a RL policy outputs the expected rewards of following that policy from a given start state to reach a given goal. Hence, the function contains information about what goals are currently solvable and what goals are not, as well as what goals are right at the cusp of being solved. To retrieve this information, we present Value Disagreement based Sampling (VDS) as a goal proposer. Concretely, we approximate the epistemic uncertainty of the value function, and then sample goals from the distribution induced by this uncertainty measure. For goals that are too easy, the value function will confidently assign high values, while for goals that are too hard, the value function will confidently assign low values. But more importantly, for the goals right at the boundary of the policy's ability, the value function would have high uncertainty and thus sample them more frequently.

To compute the epistemic uncertainty practically, following recent work in uncertainty measurement [30], we use the disagreement between an ensemble of value functions. For evaluation, we report learning curves on 18 challenging sparse-reward tasks that include maze navigation, robotic manipulation and dexterous in-hand manipulation. Empirically, VDS further improves sample efficiency compared to standard RL algorithms. Code is publicly available at https://github.com/zzyunzhi/vds.

## 2 Background and Preliminaries

Before we describe our framework, we first discuss relevant background on goal-conditioned RL. For a more in-depth survey, we refer the reader to Sutton et al. [52], Kaelbling et al. [23].

## 2.1 Multi-Goal RL

We are interested in learning policies that can achieve multiple goals (a universal policy). Let $\mathcal{S}, \mathcal{A}$ be the state space and action space as in standard RL problems. Let $\mathcal{G}$ be the parameter space of goals. An agent is trained to maximize the expected discounted trajectory reward $\mathbb{E}_{s_{0:T-1}, a_{0:T-1}, r_{0:T-1}, g}\left[\sum_{t=0}^{T-1} \gamma^t r_t\right]$, where a goal $g$ is sampled from the parameter space $\mathcal{G}$. Multi-goal RL problem can be cast as a standard RL problem with a new state space $\mathcal{S} \times \mathcal{G}$ and action space $\mathcal{A}$. Policy $\pi : \mathcal{S} \times \mathcal{G} \rightarrow \mathcal{A}$ and Q-function $\mathcal{S} \times \mathcal{G} \times \mathcal{A} \rightarrow \mathbb{R}$ can be trained with standard RL algorithms, as in [45, 3].

Following UVFA [45], the sparse reward formulation $r(s_t, a, g) = [d(s_t, g) < \epsilon]$ will be used in this work, where the agent gets a reward of $0$ when the distance $d(\cdot, \cdot)$ between the current state and the goal is less than $\epsilon$, and $-1$ otherwise. In the context of a robot performing the task of picking and placing an object, this means that the robot gets a higher reward only if the object is within $\epsilon$ Euclidean distance of the desired goal location of the object. Having a sparse reward overcomes the limitation of hand engineering the reward function, which often requires extensive domain knowledge. However, sparse rewards are not very informative and makes optimization difficult. In order to overcome the difficulties with sparse rewards, we employ Hindsight Experience Replay (HER) [3].

## 2.2 Hindsight Experience Replay (HER)

HER [3] is a simple method of manipulating the replay buffer used in off-policy RL algorithms that allows it to learn universal policies more efficiently with sparse rewards. After experiencing some episode $s_0, s_1, ..., s_{T-1}$, every transition $s_t \rightarrow s_{t+1}$ along with the goal for this episode is usually stored in the replay buffer. However, with HER, the experienced transitions are also stored in the replay buffer with different goals. These additional goals are states that were achieved later in the episode. Since the goal being pursued does not influence the environment dynamics, we can replay each trajectory using arbitrary goals, assuming we use off-policy optimization [43].

# 3 Method

We first introduce Goal Proposal Module, a module that generates an automatic curriculum for goals. Following this, we describe our Value Disagreement Sampling (VDS) based Goal Proposal Module.

## 3.1 Goal Proposal Module

Let $\mathcal{C} : \mathcal{G} \rightarrow \mathbb{R}$ be a probability distribution over the goal space $\mathcal{G}$. A goal proposal module samples a goal $g$ from $\mathcal{C}$ at the start of a new episode. In this episode, the agent follows a $g$-conditioned policy to perform a trajectory and receives external rewards defined in Section 2.1.

In standard goal-conditioned RL, $\mathcal{C}$ reduces to the uniform distribution, where the goals are randomly sampled. However, sampling goals uniformly is often uninformative for the learning process [3] since during the early stages of learning, a majority of sampled goals are too hard, while during the later stages of learning most goals are too easy. Instead of using a uniform distribution over the goals, a curriculum learning based approach can sample goals in increasing order of difficulty.

To explicitly account for the dependence of $\mathcal{C}$ on the current policy $\pi$ as normally in the case of curriculum learning, we denote the goal distribution as $\mathcal{C}^\pi$. In the case when the starting position varies, $\mathcal{C}$ could also depend on the first state of an episode $s_0$, but $s_0$ is dropped from notation for simplicity.

## 3.2 Value disagreement

To automatically generate the goal sampling curriculum $\mathcal{C}^\pi$, we propose using the epistemic uncertainty of the Q-function to identify a set of goals with appropriate difficulty. When the uncertainty for $g \in \mathcal{G}$ is high, $g$ is likely to lie at the knowledge frontier of policy $\pi$ and thus is neither too easy nor too difficult to achieve. We defer more detailed reasoning and empirical evidence to Section 4.5.

---

**Algorithm 1** Curriculum Learning with Value Disagreement Sampling

---
**Input:** Policy learning algorithm $A$, goal set $\mathbb{G}$, replay buffer $R$.
**Initialize:** Learnable parameters $\theta$ for $\pi_\theta$ and $\phi_{1:k}$ for $Q_{1:k}$
**for** $n=1,2,..N_{\text{iter}}$ **do**
    Sample a set of goals $\mathbb{G}$
    Compute $\hat{\mathcal{C}}^{\pi_\theta}$ according to equation 2
    Sample $g \sim \hat{\mathcal{C}}^{\pi_\theta}(\cdot)$
    Collect a goal-conditioned trajectory $\tau_n(\pi_\theta \mid g)$
    Store transition data into the replay buffer $R \leftarrow \tau_n$
    **for all** $\phi \in \{\phi_1, ..., \phi_k\}$ **do**
        Perform Bellman-update according to equation 1 on samples drawn from $R$
    Update policy parameter $\theta$ using algorithm $A$
**Return:** $\theta$

---

Let $Q_\phi^\pi(s, a, g)$ be the goal-conditioned Q-function of a policy $\pi$, where $\phi$ is a learnable parameter. It approximates the expected cumulative return $\mathbb{E}_{s_0=s,a_0=a,\tau\sim\pi(.|g)}\left[\sum_{t=0}^{T-1}\gamma^t r_t\right]$. Given a transition $(s, a, r, s', g)$, this Q-function can be optimized using the Bellman update rule [52]:

$$Q_\phi^\pi(s, a, g) \leftarrow r + \gamma \mathbb{E}_{a'\sim\pi(\cdot|s,g)}\left[Q_\phi^\pi(s', a', g)\right] \tag{1}$$

Intuitively, this function tracks the performance of the policy $\pi$.

In practice, to estimate the epistemic uncertainty of $Q^\pi$, we measure the disagreement across an ensemble of parametric Q-functions following Lakshminarayanan et al. [30]. Hence, instead of a single $Q$, we maintain $K$ Q-functions $Q_{1:K}$ with independently trained parameters $\phi_{1:K}$. The disagreement between value functions for a goal $g$ is computed as a function of the ensemble's standard deviation.

Formally, let $s_0, a$ be the starting state of a new episode and the agent's action. The agent's action is chosen based on the base policy optimization algorithm. For any given a goal $g \in \mathcal{G}$, let $\delta^\pi(g)$ be the standard deviation of $\{Q_1^\pi(s_0, a, g), \cdots, Q_K^\pi(s_0, a, g)\}$. Given any function $f : \mathbb{R}^* \to \mathbb{R}^*$, we define $\mathcal{C}^\pi(g) = \frac{1}{Z}f(\delta^\pi(g))$, where $Z = \int_{\mathcal{G}} f(\delta^\pi(g)) \, \mathrm{d}g$ is the normalization constant.

Since $Z$ is usually intractable, we first uniformly sample a set of goals $\mathbb{G} = \{g^{(n)}\}_{n=1}^N \subseteq \mathcal{G}$. Then we define $\hat{\mathcal{C}}^\pi : \mathbb{G} \to \mathbb{R}$ as:

$$\hat{\mathcal{C}}^\pi(g) = \frac{1}{\hat{Z}}f(\delta^\pi(g)) \tag{2}$$

to approximate $\mathcal{C}^\pi$, where $\hat{Z} = \sum_{n=1}^N f(\delta^\pi(g^{(n)}))$.

Our method is summarized in Algorithm 1. For our experiments, we use DDPG [31] as our base RL algorithm to train the policy. We define $f$ to be the identity function for simplicity for all our experiments. Ablation study on multiple choices of $f$ is deferred to Appendix E.

Note that although we use an off-policy algorithm as our base optimizer, the goal sampler is independent of the choice of the base RL optimizer. The base RL algorithm is agnostic of the value ensemble and receives training goals only via Goal Proposal Module. The value ensemble has access to transition data collected by the agent, but the base RL algorithm is treated as a black box to maintain maximum flexibility.

## 4 Experiments

In this section, we first describe our experimental setup, training details and baseline methods for comparison. Then, we discuss and answer the following key questions: (i) Does VDS improve performance?; (ii) Does VDS sample meaningful goals?; (iii) How sensitive is VDS to design choices?

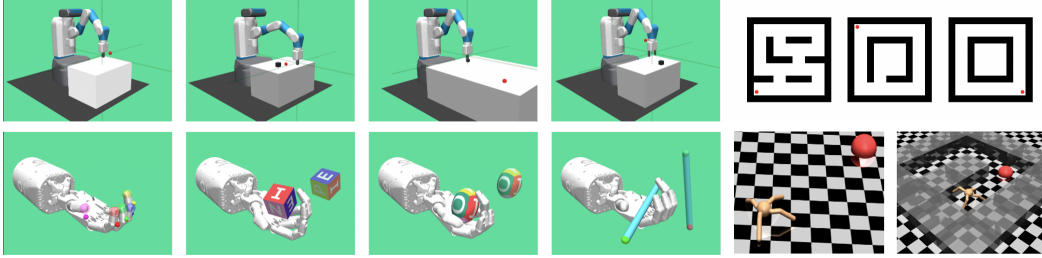

Figure 2: We empirically evaluate on all 13 robotic environments from OpenAI Gym [40], of which we illustrate 8. We also test our method on 3 maze navigation tasks, which serve as simple tasks for investigating VDS. In order to compare with GoalGAN [16], we evaluate our method on two Ant environments borrowed from their paper. The red dots, if illustrated, represent goals the robot or the object needs to reach.

## 4.1 Experimental setup

We test our methods on 13 manipulation goal-conditioned tasks, 3 maze navigation tasks and 2 Ant-embodied navigation tasks, all with sparse reward, as shown in Figure 2. Detailed setup of the environments is presented in Appendix C.

## 4.2 Training details

To enable modularity, we treat the value ensemble as a separate module from the policy optimization. This provides us the flexibility to use VDS alongside any goal-conditioned RL algorithm. In this work, we use HER with DDPG as our backbone RL algorithm. To collect transition data, the policy optimizer queries the value ensemble to compute the goal distribution and select training goals accordingly. In line with standard RL, the policy generates on-policy data with $\epsilon$-greedy strategy. The obtained transitions data are then fed into the replay buffer of the VDS's value ensemble along with the replay buffer of the policy. In each training epoch, each Q-function in the ensemble performs Bellman updates with independently sampled mini-batches, and the policy is updated with DDPG. Evaluation goals are randomly selected, and the goal is marked as successfully reached if the agent reaches the goal at the last timestep of the episode. Detailed hyper-parameter settings are specified in the Appendix D, while an analysis of combining HER with VDS is provided in Appendix G.

## 4.3 Baseline Methods

To quantify the contributions of this work, we compare our method with the following baselines:

- **HER** In HER, the RL agent uses a hindsight replay buffer with DDPG [31] as the base RL algorithm with goals uniformly sampled from the goal space. The implementation and hyperparameters is based on the official codebase of HER. We use the same set of hyperparameters for HER and our method across all environments.

- **Robust Intelligence Adaptive Curiosity (RIAC)** RIAC [9] proposes to sample goals from a continuous goal space $\mathcal{G}$ according to the Absolute Learning Progress (ALP) of the policy. The policy has large positive learning progress on a region of $\mathcal{G}$ when it is making significant improvement on reaching goals lying within region. It has negative learning progress when suffering from catastrophic forgetting. RIAC splits $\mathcal{G}$ into regions, computes the ALP score for each region, selects regions with a probability distribution proportional to the ALP score, and samples goals from the selected regions.

- **Covar-GMM** Covar-GMM [34] fits a Gaussian Mixture Model (GMM) on the goal parameter space $\mathcal{G}$ concatenated with the episodic reward and time. At the start of each episode, a goal is sampled with propability proportional to the covariance of episodic reward and time.

- **ALP-GMM** ALP-GMM [42] fits a GMM on $\mathcal{G}$ concatenated with an ALP score approximated by the absolute reward difference between the current episode conditioned on $g$ and a previous episode conditioned some goal neighboring $g$. Our implementation and hyperparameters of RandomSAC, RIAC, Covar-GMM and ALP-GMM follows the official codebase of ALP-GMM.

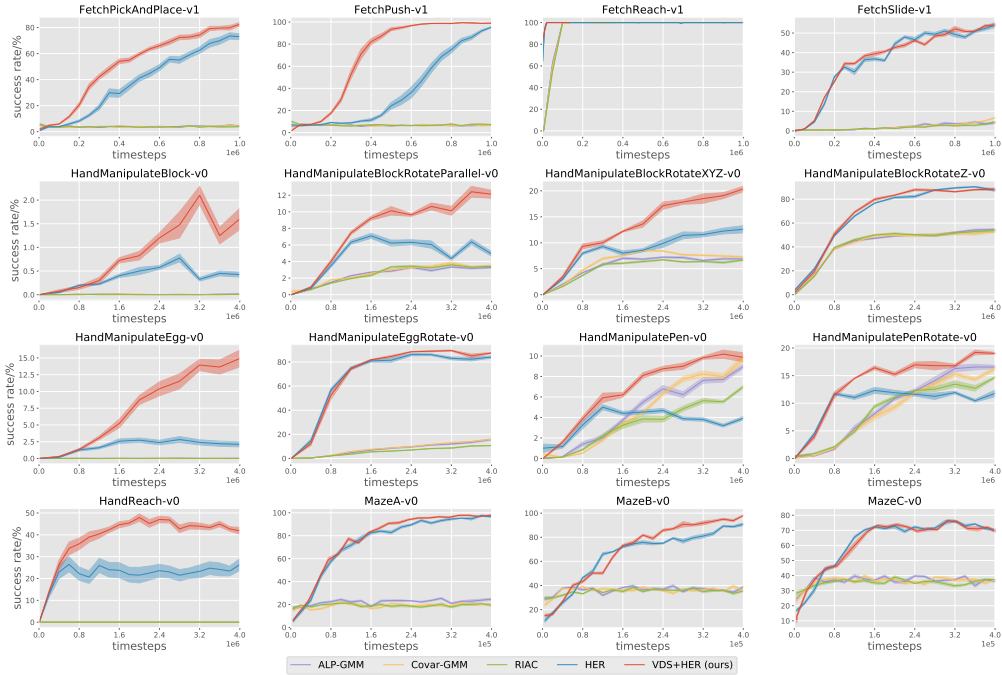

Figure 3: Here we visualize the learning curves on 16 environments that include all 13 OpenAI Gym robotics benchmark environment and the 3 Maze environments. The $y$-axis is the success rate evaluated with the latest policy. The shaded region represents confidence over 5 random seeds. We notice significant improvements in sample efficiency of our method compared to baseline algorithms, especially on many challenging manipulation tasks.

- **GoalGAN** GoalGAN [16] labels if the goals in the replay buffer are of intermediate difficulty by the episodic reward, and then feed the labeled goals into a Generative Adversarial Network (GAN) that outputs goals of intermediate difficulty. In later episodes, the agent is trained on goals generated by GAN. Our implementation and hyperparameter settings of GoalGAN follow their official codebase.

## 4.4 Improvements using VDS

Figure 3 shows that our method achieves better sample efficiency compared to baselines on most of the 16 environments with 4 FetchArm, 9 HandManipulation and 3 Maze navigation tasks. Uniform goal sampling (HER) demonstrates competitive performance in some of the reported environments, which is consistent with previous work in Portelas et al. [42].

We compare with GoalGAN in the Ant environments as reported in their original paper [16]. Figure 4 shows that our method obtains significant sample efficiency gain compared to Goal-GAN. One difference of the environment is that our method and HER perform fixed-length trajectories, and episodic success is measured as whether the goal is reached in the final timestep of the episode; in contrast, in GoalGAN, whenever the agent achieves the goal, the goal is marked as reachable and the episode terminates before reaching maximal episode length. This is consistent with the original implementation and works in favor of GoalGAN. Also note that the curve does not take into account timesteps used to label the difficulty of goals, again in favor of

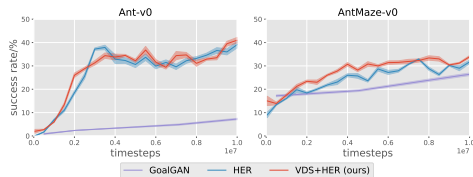

Figure 4: We compare VDS+HER, HER and GoalGAN on two ant environments. All curves are averaged over 5 seeds, with the shaded area representing confidence. $y$-axis is the evaluation success rate of the latest policy, and $x$-axis is timesteps. We show that both our method and RandomDDPG achieves better sample efficiency compared to GoalGAN.

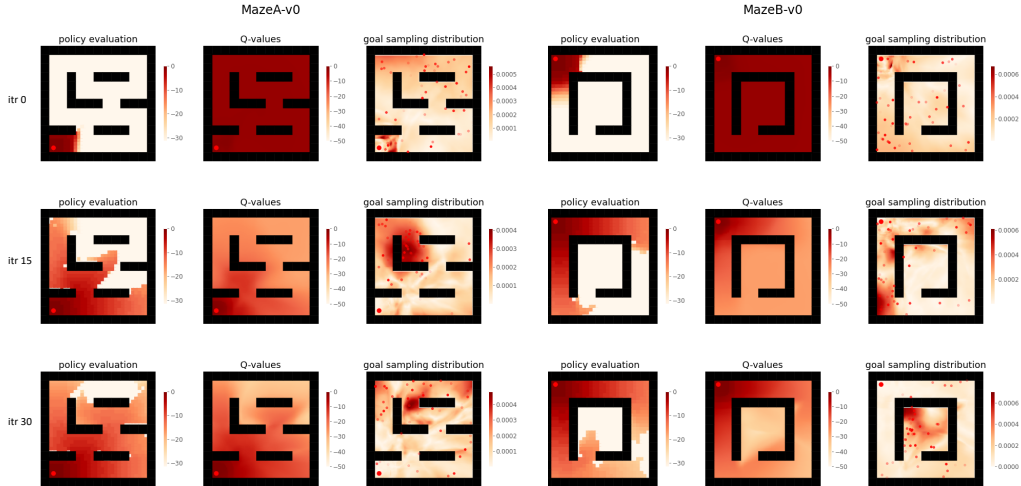

Figure 5: We illustrate the goal-conditioned episodic rewards of the latest policy, Q-values averaged over the ensemble, and finally the goal sampling distribution with sampled training goals (red dots) for two Maze environments shown in Figure 2. The agent starts from the bottom-left corner in MazeA and top-left for MazeB. We note that the disagreement produces a higher density of samples on regions at the frontier of learning. Over iterations, we also see the sampled goals move away from the starting state and towards harder goals. A complete illustration is available at https://sites.google.com/berkeley.edu/vds.

GoalGAN. We conclude from these two environments that our method is more sample efficient than GoalGAN, self-play [51], SAGG-RIAC [10], uniform sampling and uniform sampling with L2 loss in these two environments. We refer the readers to Florensa et al. [16] for performance curves of these baselines.

Learning in these environments, with the absence of strong learning signals, typically requires effective exploration. Our results demonstrate that VDS in combination with hindsight data sets up an informative course of exploration.

## 4.5  Does VDS sample meaningful goals?

To have an intuitive understanding of the goal sampling process and how VDS helps with setting up learning curriculum, we visualize in Figure 5 the followings: (i) the evaluated trajectory returns of the policy conditioned on goals varying over the maze world, (ii) Q-value predictions averaged over the ensemble, and (iii) goal distribution with VDS and the most recent 50 training goals.

In (i) and (ii), visualization of reward and Q-value landscape shows that the policy gradually expands its knowledge boundary throughout the training process. Darker region indicates areas that the policy achieves higher trajectory rewards or higher Q-values. In (iii), darker region indicates higher uncertainty of the ensemble prediction, which matches the boundary of (i) and (ii).

At the start of training, goal sampling distribution is close to uniform due to random initialization. Then, as the policy learns to reach goals neighboring to the starting position, it is also possible to reach goals residing close to the learning frontier, as minor disturbance with $\epsilon$-greedy strategy could lead the policy to hit the goal and obtain the corresponding reward signal. These goals are not yet mastered by the policy but could happen to be reached by policy exploration, and therefore have higher Q-value prediction variance. With VDS, they are more likely to be selected.

These goals at the frontier are ideal candidates to train the policy, because they are nontrivial to solve, but are also not as hard compared to goals lying far away from the policy's mastered region. Consequently, VDS improves sample efficiency by setting up an automatic learning curriculum. Figure 5 indeed suggests so, as we notice a clear sign of a goal distribution shift over iterations, with harder and harder goals getting sampled.

### 4.6 Ablations on VDS

To understand the effects of various design choices in implementing VDS (see Section 3.2), we run ablation studies. Specifically, we study the effects of (i) choice of sampling function $f$, (ii) choice of ensemble size for value uncertainty estimation, and (iii) options of combination with HER [3]. While details of these ablations are deferred to Appendix E, F, G, we highlight key findings here. First, for sampling functions, we find that our method is insensitive to the choice of sampling function $f$. Second, VDS is not sensitive to ensemble size. In fact, performance when using an ensemble size of 10 is the same as using an ensemble size of 3. Finally, we show that without using HER, VDS still improves the performance of vanilla DDPG. Combining with results in more environments from Section 4.4, we conclude that VDS is complementary with HER and provides the best result when used together.

## 5 Related Work

Our work is inspired from and builds on top of a broad range of topics across curriculum learning and goal-conditioned reinforcement learning. In this section, we overview the most relevant ones.

### 5.1 Curriculum Learning

Automatic curriculum generation has a rich history in the context of supervised learning. Bengio et al. [12] demonstrates how gradually increasing the complexity of training samples to a supervised learning algorithm leads to accelerated learning and better prediction quality. Kumar et al. [29] then proposed 'self-paced learning' in the context of non-convex optimization, where the order of training examples is automatically chosen. Murali et al. [35] demonstrates how automated curriulums on the control space can improve performance of robotic grasping. In all of these works, the focus is on supervised learning problems, where the curriculum is over training examples that are fed to the learning algorithm. Our work builds on top of this idea to creating curriculums over tasks/goals that a RL algorithm needs to solve.

In the context of decision making problems, several techniques to generate curriculums have been proposed. Graves et al. [19] demonstrates that having a curriculum accelerates learning in multi-armed bandit settings, and Matiisen et al. [32] proposes a Teacher-Student Curriculum Learning (TSCL) framework. Both works focus on discrete goal space instead of continuous goal space in most standard goal-conditioned RL environments. Pong et al. [41], Racaniere et al. [44] propose curriculum learning algorithms for pixel-based environments with no prior access to uniform sampling in the goal space due to high dimension of visual inputs, while in this work we focus on environments with state-based observations.

Florensa et al. [17] has a problem setting more similar with ours. It proposes reverse curriculums, where given a specific goal to solve, the agent is reset to a states closer to the goal and then over time expanded. However, this assumes easy reset to arbitrary states, which is not practical for general purpose RL. To alleviate this, HER [3, 1] samples goals based on states previously reached by the agent using 'hindsight'. As the agent improves performance, the state footprint of the policy increases and hence more complex goals are sampled. However, using this strategy a large portion of the sampled goals are too easy to provide useful signal. In our work, we combine VDS with HER and show significant improvements over vanilla HER. Several recent works have looked at creating curriculums by explicitly modelling the difficulty of the goal space [9, 34, 42, 16]. Again, we show empirically that VDS obtains substantial performance gains over previous automatic curriculum techniques (see Section 4.4 and Figure 3).

### 5.2 Self-Play based curriculums

Sampling tasks/goals that are useful for RL has also been studied in the context of 'self-play' [14], where a two-player competitive game is setup in which different player policies are pitted against each other. This technique has seen success in challenging games like GO [49] and DOTA [13]. In the context of robotic control problems, Bansal et al. [8] demonstrates how self-play can assist in the development of locomotion behaviors. Instead of a symmetric game setting, Pinto et al. [39], Sukhbaatar et al. [51] propose setting up asymmetric two-player games, where one agent focuses on proposing goals without having to explicitly solve that goal. These self-play setting create

an automated curriculum that improves learning. However, applying these ideas to arbitrary control problems, where a natural game formulation is not present, is challenging. Recently, GoalGAN [16] has shown superior performance to such asymmetric game settings. Empirically, both VDS and HER perform significantly better than GoalGAN on Ant-navigation tasks (see Section 4.4 and Figure 4).

## 5.3 Uncertainty estimation

Uncertainty estimation has been widely used for exploration in decision making problems such as multi-armed bandits and active learning[47], with classic approaches including Thompson Sampling [53] and UCB algorithm[5]. More recent works have looked at estimating uncertainty of a function approximator [6, 27, 18, 30, 36, 11], which is a key component in our method. Since we are looking to estimate the epistemic uncertainty, i.e. the prediction uncertainty of a model, we use an ensemble of neural networks inspired from Lakshminarayanan et al. [30]. This use of uncertainty has been previously applied for exploration [21, 37], and although we draw architectural inspiration from these works, we note that our problem setting is different.

## 6   Conclusion

In this work we present a technique for automatic curriculum generation of goals that relies on the epistemic uncertainty of value functions. Through experiments on a suite of 18 sparse-reward tasks, we demonstrate substantial improvements in performance compared to HER and other standard RL algorithms. Through further analysis, we demonstrate that our method is robust to different hyperparameters while being able to sample goals at the frontier of the learning process. Finally, we believe that this simple technique can be extended to other domains like real-world robotics and visual reinforcement learning.

## Broader Impact

While recent advancements in Artificial Intelligence (AI) provide a lot of potential opportunities to create products such as in elderly care, medical consultant and surgery assistant, autonomous devices that keep the supply chain sustainable under extreme working conditions or during pandemic, etc. It also provides tools that utilize the availability of large amount of data today, with the end-goal of improving our quality of life. There are however multiple possible negative consequences that we must be aware of: (i) training neural networks is typically associated with large energy consumption that is harmful to the environment, (ii) the significant computation resource requirement prevents many researchers from accessing state-of-the-art research, and (iii) in the field of RL in particular, algorithms typically require a long duration of interaction with the environments, which could further introduce a prohibitive monetary cost when deployed on physical robotic environments. In this paper, we focus on improving sample efficiency when training a universal agent that can perform a range of tasks. Furthermore, VDS is accessible to a broad range of researchers (even those without access to GPUs) and leaves a much smaller carbon footprint than competing methods. In fact, all of our experiments using VDS are run on a single CPU.

It is fair to say that even with the result of this paper, Deep RL agents are far from being applicable to complex problems in real life and being widely accessible to the public. Regardless, we believe this paper provides progress and contributes to the goal of making reliable robotic applications. Along this process we are aware that evaluating robot safety is a crucial part of the consideration. Therefore rather than exclusively focusing on developing state-of-the-art algorithms, we draw attention to complementary research on safety [2, 22].

## Acknowledgments and Disclosure of Funding

We gratefully acknowledge the support Berkeley DeepDrive, NSF, and the ONR Pecase award. We also thank AWS for computational resources.

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
