[Supplementary Material]

## Appendix

## A    Reinforcement learning

In our continuous-control RL setting, an agent receives a state observation $s_t \in \mathcal{S}$ from the environment and applies an action $a_t \in \mathcal{A}$ according to policy $\pi$. In our setting, where the policy is deterministic, we hence have $a_t = \pi(s_t)$. The environment returns a reward for every action $r_t$. The goal of the agent is to maximize expected cumulative discounted reward $\mathbb{E}_{s_{0:T}, a_{0:T-1}, r_{0:T-1}} \left[ \sum_{t=0}^{T-1} \gamma^t r_t \right]$ for discount factor $\gamma$ and horizon length $T$. On-policy RL [46, 24, 54] optimizes $\pi$ by iterating between data collection and policy updates. It hence requires new on-policy data every iteration, which is expensive to obtain. On the other hand, off-policy reinforcement learning retains past experiences in a replay buffer and is able to re-use past samples. Thus, in practice, off-policy algorithms have been found to achieve better sample efficiency [31, 20]. For our experiments we use DDPG [31] as our base RL optimizer due to its sample efficiency and fair comparisons with baselines that also build on top of DDPG. However, we note that our framework is compatible with any standard off-policy RL algorithm.

## B    Deep Deterministic Policy Gradients (DDPG)

Deep Deterministic Policy Gradients (DDPG) [31] is an *actor-critic* RL algorithm that learns a deterministic continuous action policy. The algorithm maintains two neural networks: the policy $\pi_\theta : \mathcal{S} \to \mathcal{A}$ (with neural network parameters $\theta$) and a $Q$-function approximator $Q_\phi^\pi : \mathcal{S} \times \mathcal{A} \to \mathbb{R}$ (with neural network parameters $\phi$). During optimization, episodes collected using $\pi$ are stored in a replay buffer. Then, the $Q$-function is optimized by minimizing the one-step Bellman error on samples from the replay buffer, while the policy is optimized using the deterministic policy gradient [48].

## C    Environments

- **Fetch:** These environments simulate a 7-DoF Fetch arm, with 3-dimensional goal space and 4-dimensional action space. The reaching task has 10-dimensional observation space, while other tasks involve objects and have 25-dimensional observation space. The agent receives a reward of 0 if the final position of the gripper of object, depending on the environment, is within $\epsilon$ Euclidean distance of the goal, and -1 otherwise. The initial position of the Fetch arm is not guaranteed to be close to the object location and henceforth increase the complexity of the task. Moreover, since the robot has to perform multiple primitive actions like grasping followed by reaching, learning this with sparse rewards is additionally challenging.
    - **FetchReach:** Move the gripper to a target location.
    - **FetchPickAndPlace:** Pick up a box and move to a target position.
    - **FetchPush:** Push a box to a target position.
    - **FetchSlide:** Slide a puck to a target position, which is outside the arm's reach.
- **Hand:** These environments simulate a 24-DoF robotic hand, with 20-dimensional action space. The reach task has 15-dimensional goal space and 63-dimensional observation space, and the manipulation tasks have 7-dimensional goal space and 61-dimensional observation space. The large action space of coupled with complex dynamics and sparse rewards makes these tasks challenging.
    - **HandManipulateBlock:** Rotate a block to match a target rotation in all axes and to match a target position.
    - **HandManipulateBlockRotateParallel:** Rotate a block to match a target rotation in $x$- and $y$-axis.
    - **HandManipulateBlockRotateXYZ:** Rotate a block to match a target rotation in all axes.
    - **HandManipulateBlockRotateZ:** Rotate a block to match a target rotation in $z$-axis.

- **HandManipulateEgg:** Rotate an egg to match a target rotation in all axes and a target position.
- **HandManipulateEggRotate:** Rotate an egg to match a target rotation in all axes.
- **HandManipulatePen:** Rotate an pen to match a target rotation in all axes and a target position.
- **HandManipulatePenRotate:** Rotate an pen to match a target rotation in all axes.
- **HandReach:** Move to match a target position for each finger tip.

- **Maze:** The environment for navigation tasks is a finite-sized, 2-dimensional maze with blocks. The agent is given a target position and starts from a fixed point in the maze, and it obtains reward of 0 if it gets sufficiently close to the target position at the current timestep, or a penalty of -1 otherwise. The agent observes the 2-D coordinates of the maze, and the bounded action space is specified by velocity and direction. The agent moves along the direction with the velocity specified by the action if the new position is not a block, and stays still otherwise. The maximum timestep of an episode is set to 50.

  - **MazeA:** The first variant of maze contains random blocks as shown in the left maze Figure 2.
  - **MazeB:** The second variant of maze is shown in the middle maze Figure 2.
  - **MazeC:** The third variant of maze is shown in the right maze Figure 2. The central area is infeasible as all sides are blocked by walls.

## D   Implementation Details and Hyperparameters

For all DDPG-based methods, we run with 1 CPU and two parallel environments. Each Q-function in the ensemble is trained with its target network, with learning rate 1e-3, polyak coefficient 0.95, buffer size 1e6, and batch size 1000. For DDPG and HER, all hyperparameters inherit from the official implementation of HER. The learning rate is 1e-3, polyak coefficient is 0.95, buffer size is 1e6, batch size is 256, the $\epsilon$-exploration coefficient is 0.3, and the standard deviation of Gaussian noise injected to non-random actions is 0.2 of maximum action.

For SAC-based methods, we run with 1 CPU and a single environment. The policy, Q-function and Value function are trained with learning rate 1e-3, buffer size 2000000, batch size 1000. All hyperparameters inherit from the official GMM implementation.

For all implementations unless stated otherwise, we employ HER. The expected ratio of transitions with swapped goal to regular transitions is 4 in each batch.

## E   Ablation: How important is the choice of disagreement function

There could be multiple choices of the disagreement function that maps from value ensemble variance to disagreement, and the value of disagreement will be subsequently fed into the goal proposal module. As mentioned in Section 3.2, we use the value ensemble variance for simplicity across all experiments. However, any monotonically increasing function $f$ that takes in ensemble standard deviation as input could be valid candidates. The choice of $f$ either smooths the goal distribution, or aggregates more weight on a small region of goals. One extreme case will be using a constant function. In this case, VDS is reduced to uniform goal sampling.

In this section, we compare the effect of different choices of $f$. Examples are exponential function $f(\delta^\pi(g)) = \exp(\delta(g))$, tanh function $f(\delta(g)) = \tanh(\delta(g))$, and square function $f(\delta(g)) = \delta^2(g)$. Results shown in Figure 6 indicate that our method is not sensitive to the choice of $f$.

## F   Ablation: How important is the choice of ensemble size?

In our experiments, we use an ensemble size of 3 for all tasks. But what happens when we vary the ensemble size? Results in Figure 7 demonstrate that the size of the value ensemble does not have a significant impact on the performance for the maze navigation task. Thus, with just a few networks in the ensemble, we can obtain performance similar to larger ensemble networks. The robustness to ensembles also points towards the fact that extremely accurate estimates of epistemic uncertainty is

Figure 6: All curves for the ablation study are ran across 5 random seeds, and the shade represents the confidence. Here we compare the effects of using different disagreement sampling strategies on the Maze environments. We notice that our method is not sensitive to the choice of $f$.

Figure 7: We compare the effects of the ensemble size used in the computation of value disagreement. We notice no degradation in performance, which points towards robust estimation of epistemic uncertainty.

Figure 8: Here we compare different method of applying VDS and HER on vanilla DDPG. We note that our method with no HER (purple) performs better than vanilla DDPG (red).

not required for seeing benefits of our approach. An additional benefit of this robustness is that our algorithm should be easily compatible with other techniques to measure epistemic uncertainty like Dropout [18].

## G    Ablation: How does VDS perform in combination with HER?

Our method can be combined with HER in various ways. Curves in Figure 8 correspond to different combinations of using or not using VDS, with and without hindsight experience data. Applying VDS on vanilla DDPG improves the performance, while only applying HER is already competitive in these twow environments. However together with results in Section 4.4 where VDS improves the sample efficiency of HER in a majority of environments, it shows that it produces the best result when using our method in combination with HER.

While HER focuses on obtaining learning signal via relabelling without extra interaction with the real environment, our method appeals to the value ensemble to conduct the interaction with the real environment in an efficient manner. VDS as a sample collection strategy could be considered as orthogonal to the way that HER improves with training, and the two methods collaboratively strengthen the learning signal that the policy processes throughout training.