[Reviews · NeurIPS 2020]

Review 1

Summary and Contributions: The paper focuses on multi-goal sparse-reward reinforcement learning, and proposes a method to increase sample-efficiency through curriculum learning: the agent learns several Q-Functions, and their disagreement (standard deviation of their predicted Q-Values in the paper) is used to measure the confidence of the agent in a particular state. This confidence can be mapped to a "how much is this state at the edge of what the agent can do" measure. Then, this measure is combined with Hindsight Experience Replay to allow the agent learn as if a nice sequence of progressively-harder goals had been given to it.

Strengths: The paper is very well-written and flows perfectly. The proposed approach is sound, elegant, and generalizes to any reinforcement learning algorithm compatible with hindsight experience replay. A good property of the proposed algorithm is that it allows the agent to learn efficiently, as it received a curriculum of states, but does not assume the environment to be resettable to any state. This, along with sample-efficiency, is crucial for real-world tasks. The empirical evaluation is thorough and shows highly encouraging results.

Weaknesses: A small weakness is that the uncertainty measure being used (the disagreement between Q-Functions) may be a bit weak or imprecise in some cases. Figure 5, for instance, illustrates that the disagreement function does not seem to change shape that much as learning progresses. It is far from a nice and clean border that progressively moves away from the original state. I believe that a higher-quality uncertainty estimate may allow the agent to receive an even better curriculum. The problem may come from the fact that neural networks do not always behave properly and predictably in unseen states (or not-often-visited states). In a collection of neural networks, some of them may all learn the same values (for instance 0, or the average Q-Value of all the states, as if the agent was a Bandit), while some of them may extrapolate to values that become arbitrarily large or small when moving away from the known region of the state-space. It is therefore difficult to know, and it may change every time-step, whether the standard deviation of predicted Q-Values will be high or low in unknown regions of the state-space. This phenomenon is sometimes discussed in the Intrinsic Motivation RL literature, when disagreement is used to compute exploration bonuses, or density estimates (discussed in [1], papers it cites, and papers that cite it). [1]: Bellemare, Marc, et al. "Unifying count-based exploration and intrinsic motivation." Advances in neural information processing systems. 2016. Author response: the authors identify the choice of uncertainty measure as a potential future work area. I therefore see this paper as having as main contribution "map an uncertainty measure to hindsight goals", which is quite interesting. The paper then proposes to use multiple Q-functions as an example of how an uncertainty measure can be computed. This is a nice story. However, I agree with the other reviewers that, because this paper touches on many topics, there are many papers to cite and discuss. This is also very important to me.

Correctness: The empirical evaluation validates the claims in the paper and seems fair. A wide range of algorithms are compared on a wide range of environments, on several runs, which follows the best practices in the field.

Clarity: The paper is very well-written, easy to follow, and contains the right amount of background information.

Relation to Prior Work: There are a few citations about intrinsic motivation reinforcement learning, and nothing important seems to be missing, but a few papers such as the following would have been welcome: [1]: Bellemare, Marc, et al. "Unifying count-based exploration and intrinsic motivation." Advances in neural information processing systems. 2016. [2]: Osband, Ian, et al. "Deep exploration via bootstrapped DQN." Advances in neural information processing systems. 2016. (also uses several Q-Functions, but does not explicitly measure the uncertainty of the agent)

Reproducibility: Yes

Additional Feedback:


Review 2

Summary and Contributions: An automatic curriculum learning algorithm is proposed to generate goals in goal reaching tasks. Inspired by prior work on uncertainty estimation, the proposed method forms goal curricula based on the disagreement of multiple trained value functions. Experiments are conducted on multiple goal-reaching robotic tasks in simulation. In most cases, performance improvements can be observed comparing with baselines. Update: After carefully reading the other reviews and the rebuttal, I still recommend rejection. I really like the writing and the algorithm design of this submission. Using value disagreement could potentially be an interesting and promising idea. However, a good research paper, in my humble opinion, should identify their intellectual contributions in the context of prior work and discuss/analyze the experimental results, which are unfortunately unsatisfying in this submission. I believe this paper is not ready for publication in its current form but has a lot of potentials if given another iteration. We had an in-depth discussion among the reviewers and I summarize the main comments/suggestions here: - As pointed out by all reviewers, this submission is missing not a few but a number of important references. This includes some of the most widely known early works in this area and a few state-of-the-art methods. The contribution of this paper cannot be convincingly justified without discussing these references in detail. The introduction section of this submission needs to be adjusted by incorporating these discussions and properly recognizing the contributions of prior work. - The motivation behind using value disagreement is not discussed in this submission, which makes their main contribution of the algorithm design seem to be heuristic and empirical. In prior work (e.g. the original Teacher-Student Curriculum Learning paper, Setter-Solver, and ALP-GMM), several learning progress indicators have been proposed such as improvement of reward, absolute learning progress, goal coverage, goal validity, and goal feasibility. While the main contribution of this work is to use value disagreement as a learning progress indicator in the Teacher-Student Curriculum Learning framework, the comparison (through either discussions or quantitative evaluations) with several of these previous works is missing. We asked for intuitions and theoretical motivations for using value disagreement. However, the authors replied that it is only supported by their qualitative results shown in Figure 5 in retrospect. But the authors did not provide any support of this claim, either from a reference or a theoretical proof in the submission or the rebuttal. - The empirical results also seem to be insufficient in several ways: 1) Several baselines that use the same high-level framework (i.e. Teacher-Student Curriculum Learning) but different learning progress indicators are not compared. 2) The performance improvement is marginal in many of the evaluated tasks. Although in the rebuttal the authors argue that the performance is better in three harder tasks, it is unclear to me what aspects of the “hardness” of these three tasks can be better tackled by the proposed method. For instance, it is especially unclear that, between the two very similar tasks, why HandManipulateEgg-v0 is considered a “hard task” (in which the proposed method is 3X better ) while HandManipulateEggRotate-v0 is not (in which the baseline has almost identical performance with the proposed method). There is no explanation in the original submission or the rebuttal. 3) In my humble opinion, strong experimental results are supposed to involve not only reporting good numbers/curves but also solid analysis and discussions based on the results. In this submission, the analysis of the results is scarce. Baselines like Covar-GMM and ALP-GMM are only briefly described with their results plot the result in Fig.5. There is not any discussion or analysis about why value disagreement should be better than the absolute learning progress used in ALP-GMM or what challenges faced by Covar-GMM and ALP-GMM can be better handled by the proposed method, in Introduction, Related Work, or Experiments. Given that the RL methods that are based on exploration strategies and curriculum learning can often be sensitive to hyperparameters and specific implementations, these empirical results in this submission can be hardly convincing without (either theoretical explanations or) solid analysis.

Strengths: - The idea of using value disagreement to form a curriculum has been widely used to encourage state visitation recently, but not in generating curriculum. The proposed idea is reasonable and interesting. - The proposed method is evaluated on a number of experiments. In more than half of the task environments, performance improvement can be obviously observed. - An ablation study is included in the paper to justify the parameter choices of the algorithm. - The writing of the paper is clear and easy to follow.

Weaknesses: - Why using value disagreement instead of other indicators of the learning progress proposed in prior work is not convincingly explained. The high-level curriculum learning framework in this paper is the same with several previous papers (e.g. Teacher-Student Curriculum Learning, Setter-Solver framework, etc.) while the main difference is using value disagreement to form the curriculum. However, a variety of indicators of learning progress have been proposed in recent works such as task feasibility, goal diversity, task similarity, etc. Without a detailed justification, the proposed solution seems to be heuristic and the contribution is not convincing. - A number of important and related works are not even mentioned in this paper. This includes the first papers which proposed automatic curriculum learning and the recent papers which proposed very similar frameworks. A few examples include but are not limited to: * Graves et al. Automated Curriculum Learning for Neural Networks. In ICML 2017. * Matiisen et al. Teacher–Student Curriculum Learning. In IEEE Trans. on Neural Networks and Learning Systems 2017. * Pong et al. Skew-Fit: State-Covering Self-Supervised Reinforcement Learning. In * Racaniere and Lampinen et al. Automated Curricula Through Setter-Solver Interactions. In ICLR 2020. Without properly recognizing with these related works and comparing with more curriculum learning and exploration baselines, the novelty of this paper cannot be convincingly justified. - The performance improvement of the proposed method seems to be incremental in many tasks shown in the paper and the experimental results are not properly discussed. In 6 of the 16 tasks in Figure 3 (e.g. FetchSlide-v1, HandManipulateEggRotate-v0, MazeA-v0, ......) and both tasks in Figure 4, the difference between the proposed method and the Random-DDPG baseline is hard to tell. It seems that by randomly sampling the goals, which is the one of the most naive baselines, the performance can already be improved. On the contrary, automatic curriculum learning baselines from prior work made the performance even worse. And these results are not discussed or explained in the main paper. - Although it is nice to see a qualitative result as shown in Figure 5, it is unclear how the goal curricula generated using the proposed method is those in different previous works. Especially given that Figure 5 is very similar to what was shown in Figure 2 in Skew-Fit. - Some of the implementation details are not described in the main paper or the supplementary material, such as the learning rate and the optimizer.

Correctness: I didn't find any false claims. However, the contributions of prior work need to be better recognized and these claims could have been better supported by the experimental results.

Clarity: Yes, the writing of the paper is clear and easy to follow.

Relation to Prior Work: As described in the weaknesses, a number of important and related works are not even mentioned in this paper. This includes the first papers which proposed automatic curriculum learning and the recent papers which proposed very similar frameworks.

Reproducibility: Yes

Additional Feedback: I would suggest the author to have a more thorough literature review of the field and rethink about the novelty of this paper as compared to prior work. It would be better to include further discussions about why using value disagreements instead of other indicators of learning progress. Comparison with more recent baseline methods should be included in terms of quantitative performance and qualitative results.


Review 3

Summary and Contributions: # UPDATE After authors rebuttal, I gained a much better understanding of authors methodology. With each Q function being trained on separate batches of data, coming from stochastic policy, the disagreement can be seen as a form of epistemic uncertainty estimation, and as such is a relatively standard, and established signal to be using for curriculum building. Consequently my previous concerns about lack of grounding are not justified. I would encourage authors to think if there is a way to incorporate this motivation into the final text, so it helps other readers to get to this conclusion easily. I still believe it is a good submission, and should be accepted (7). # REVIEW In this paper authors tackle the problem of adaptive sampling of goals in the reinforcement learning problem (i.e. curriculum learning). Authors propose to use variance of predictions of an ensemble of Q functions to drive the goal selection. The main contribution is said simple algorithm, together with its empirical validation on the suite of 13 problems from OpenAI Gym robotics and 3 maze environments. Presented results show consistent improvements over the baseline method of sampling goals randomly. *After reading the rebuttal, I still recommend acceptance (7) at a high confidence level (4)*

Strengths: Authors present a very simple (conceptually and implementation-wise) method of automatically building curriculum over goals in reinforcement learning. Empirical evaluation looks solid, despite being limited to specific kind of RL tasks. Topic covered is very relevant for both NeurIPS community and general RL audience, as automatic building of curriculum, open ended learning regimes, are one of the hypothesised paths towards scaling up RL to real world problems.

Weaknesses: Proposed method is driven by disagreement between value functions. However, nothing on the mathematical level drives such disagreement to exist, nor to be related to learnability or value for the policy. While the experiments in the space considered seem to help, currently method lacks any more theoretical motivation, and it clearly has degenerate fixed points e.q.: - if all Q functions are the same, they will never stop being the same; since all Q functions are trained in the same way, as time progresses they will become more and more similar and thus the strength of the method relies purely on initial randomness [*This comments no longer applies, and was addressed by the rebuttal, where authors say that each Q function trains on separate mini batch*] - agreement between Q does not mean agent is good at a goal/task and yet it will be downweighted. This is by far the biggest issue right now, while it is akin to uncertainty sampling from active learning literature, the critical difference is that uncertainty of Q is not the same as uncertainty of \pi. While these can be correlated, they can also be completely opposite forces. - it is unclear how the method would behave in more complex environments, that involve significant sources of randomness inherit to the world. - It is unclear whether baselines used (e.g. GoalGAN, ALP-GMM) do use exactly the same RL part, based on DDPG, or authors replicated entire codebase of the baselines. It makes it hard to disentangle effects of VDS from pure RL effects (e.g. see how big gap one obtains between RandomDDPG vs RandomSAC). Arguably the correct evaluation comparison requires exactly the same RL setup, and the only difference coming from sampling g.

Correctness: Paper, as presented, seems to make correct statements about method and contribution. However, as mentioned in the weakness section it is hard to assess method correctness itself as it does not seem to be really grounded, and has said degenerated fixed points.

Clarity: Paper is easy to follow, presented in clear terms, with all objects defined. The only presentation issue is scale of numbers in figures - it is impossible to read these in the printout. If numbers are supposed to be read - they should be enlarged. If they are meaningless - removed.

Relation to Prior Work: Authors provide relation to prior work, however the whole body of work on active learning (and in particular uncertainty sampling) has been omitted. I would encourage authors to include this chunk of literature in the future version of the paper, as it could help further place the research in existing body of work, and become a helpful development direction for other researcher less familiar with active learning literature (where uncertainty sampling is the first, simplest method used).

Reproducibility: Yes

Additional Feedback:


Review 4

Summary and Contributions: The paper proposes a new curriculum learning strategy for learning goal conditioned policies in environments with sparse rewards. Specifically, the authors propose using an ensemble of Q-value approximation models and sampling new goals according to a distribution defined by some function of the standard deviation of the ensemble predictions. These goals are used to generate new training trajectories, and the algorithm is employed alongside Hindsight Experience Replay (HER) to populate a replay buffer of state transitions for training. The authors experiment with the proposed method on 18 goal-conditioned simulated robotics tasks and demonstrate improved sample efficiency on some of these compared to a range of existing methods from the literature. The authors also include some exploration into the proposed method’s behavior and its agreement with motivating intuition.

Strengths: The authors described an interesting approach for goal sampling for curriculum learning of goal-conditioned RL agents and conducted a thorough study with an array of alternative methods on a wide range of environments. The paper is easy to read overall and includes some useful inspection into the method’s behavior on a simple illustrative example.

Weaknesses: The paper’s primary weakness is that the proposed method does not appear to be a sufficient improvement over the evaluated baselines to warrant wide attention from the NeurIPS community. Though negative or neutral empirical work is also valuable, the community would probably expect a more thorough empirical evaluation to warrant publishing. Specifically: 1. One might expect to see some attempt to evaluate why the new method yielded gains on some tasks but not others. Are there features of an environment which makes the problem more amenable to this approach? 2. Neutral / negative results are more interesting when the method is simple, but some decisions about this approach seem arbitrary. Most notable is the coupling with H.E.R. but one might also ask if standard deviation is the most appropriate measure of uncertainty. 3. In a more experiment-focused paper, the ablation studies are more important, but the ablation studies in the submitted paper are incomplete and (in the case of the H.E.R. ablation) are either missing or poorly explained. If I’m mistaken and this really does represent a notable improvement, the case would be stronger with at least a more concrete claim with a bit more statistical rigor.

Correctness: The experimental methodology is mostly encouraging, but I worry that among 18 tasks the new method might outperform on some of these by simple chance. The error bars from five repeat evaluations are nice, but I don’t expect this is enough for a definitive statistical claim (though to be fair, I did not see such a claim in the paper).

Clarity: The paper is well organized and generally easy to understand. Some specific instances of poor clarity include: * The intuition provided to motivate the proposed method is appreciated but was too ambiguous to be useful for me. By not specifying what is meant by a challenging task, I can’t quite agree with the premise that high epistemic uncertainty is indicative of an intermediately difficult goal. * The main text claims that an “aggressive exponential of standard deviation” performed best, but the corresponding ablation study in the appendix reports no discernible difference.

Relation to Prior Work: The authors list related work which motivated curriculum learning broadly and its role in RL, and they compare their method to Hindsight Experience Replay. Perhaps notably missing is any discussion of exploration policies, which have a long history of incorporating uncertainty (e.g. Thompson Sampling, UCB).

Reproducibility: Yes

Additional Feedback: * The abstract claims to experiment on 18 tasks, but the main text and appendix only list 16. Is this a typo? * I’m not an expert with goal-conditioned RL, so I should ask why is RandomDDPG so successful relative to what appears to be a wide array of more sophisticated methods from the literature? Are these alternate methods stronger in different environments? Is that worth noting for the reader?

[Author Response · NeurIPS 2020]

We thank the reviewers for their detailed and comprehensive review. We are glad that the reviewers have found our algorithm "elegant"(R1) and "simple"(R3), with "'solid empirical evaluations"(R1) and "highly encouraging results"(R3). However, the reviewers have raised concerns particularly regarding comparisons to existing literature (R2, R4). We address the most salient points of feedback below, and will incorporate all feedback in our paper.

—————————————————————————— **General Feedback** ——————————————————————————

**Significance of results:** In this work, we propose a simple automatic curriculum technique (VDS) in the goal-conditioned RL setting that samples goals according to the epistemic uncertainty of learned value functions. The simplicity of this method allows us to use it on a range of domains from simple ones like maze navigation to challenging ones such as multi-fingered dexterous manipulation. In an extensive evaluation on 18 domains, including **all** 13 OpenAI Gym Robotics tasks, we show that VDS is significantly better than state-of-the-art baselines such as HER on 10/18, while being on par on the remaining 8. Our results are hence not cherry-picked on domains. We are excited to see R1 and R3 both acknowledge the significance of our empirical evaluation. However, R2 and R4 do not see this as a sufficient improvement. Hence, to reiterate the significance of our results, we note that in the three hardest tasks: HandManipulateEgg, HandManipulateBlock, and HandManipulateBlockRotateParallel, we achieve $\approx 3\times$ the performance of HER, our strongest baseline. While, on the three easy Maze tasks we report no improvements. This indicates that VDS offers more value in difficult domains, without hurting performance on simpler ones.

**Acknowledgement of prior work:** We thank all four reviewers for their suggestions of prior work notably in intrinsic motivation RL (R1), curriculum learning (R2), active learning (R3), and uncertainty estimation (R4). Indeed, our work although simple, is connected to large bodies of prior work, which we will cite and discuss thoroughly in our paper. However, R2 believes that some papers we missed citing hurts the novelty of our work. Particularly, R2 cites 5 papers [a-e], and we describe why although important works, most would not be valid baselines for VDS. Graves et al. 2017 [a] and Matiisen et al. 2017 [b] is an overarching framework that explicitly tracks learning progress for every 'goal'. This is only possible to do efficiently with a small discrete goal space, and would not scale to a large continuous goal space that is used in standard goal-conditioned RL. Sukhbaatar et al. 2018 [c] is in fact included in our citations (please see Section 5.2 and [44]) and a baseline for Florensa et al. 2018, which we significantly improve upon in Fig. 4. Pong et al. 2020 [d] and Racaniere and Lampinen et al. 2020 [e] presents exciting approaches for automatic curricula with a focus on pixel-space observation and as such not directly applicable as a baseline. However, we believe both methods can be used in conjunction with VDS for better learning on image-based domains.

—————————————————————————— **Algorithmic / Experimental details** ——————————————————————————

**Why value disagreement?** (R1, R2, R4). We highlight that value disagreement assigns high probability to goals associated with high learning progress (Fig. 5.) and goals of intermediary difficulty, which are two properties that are explicitly optimized for by some previous methods. However, there are several techniques to estimate epistemic uncertainty such as using Dropout, or using Bayesian neural networks. Studying the effects of such choices and the behavior that emerges would serve as an exciting avenue for future research.

**Diversity in ensemble training** (R3). The ensemble is trained with random initialization and independent mini-batches. Thereafter, the method also benefits from the agent's exploration strategy or a stochastic RL policy. Even if all Q functions are initialized as the same, since the agent does not behave deterministically, members in the ensemble would see different transitions and will not be identical throughout the training.

**Hard goals are down-weighted** (R1, R3). Selecting a goal that is particularly difficult to reach provides little to no reward signal in a sparse-reward environment. VDS empirically down-weights these difficult goals early on in training and only begins to sample them once the policy is performative on easier goals (Fig. 5).

**Definition of a challenging task** (R4). By "challenging tasks" we refer to goals that are far away from the solvable region. The agent is unlikely to obtain reward signals with these challenging training goals given sparse reward. VDS assigns less probability to these challenging training goals since the agent consistently obtains low reward.

**Disentangling effects of VDS** (R3, R4). We thank the reviewers for their questions on clarity of baselines, which we will improve in the paper. GoalGAN is based on TRPO, while the GMM baselines use SAC. To avoid potential performance downgrade, we replicated the entire codebase of the baselines in a single framework. We note that the performance of baselines in our framework that uses the newer SAC backbone is higher than the original implementations. All our implementations will be publicly released for others to compare with and build on. Additionally, one of the baselines "RandomDDPG" incorporates HER which can bring significant benefits in sparse-reward, goal-conditioned tasks, and hence in certain domains performs better than more recent curriculum learning baselines (Fig. 4).

**Number of environments** (R4). The 18 tasks include 16 tasks in Fig. 3 and 2 Ant environments in Fig. 4.

We thank the reviewers for their thorough and thoughtful review. Due to constraints in space, we had to exclude discussions on interesting questions, and we will defer detailed analysis to the main paper.

[Meta-Review · NeurIPS 2020]

This paper tackles the problem of adaptive goal sampling to automate curriculum learning by using value disagreement of an ensemble of models as a proxy. The method is clearly motivated and explained, with a wide set of experiments showing the mechanics of the method working in the intended way, and improvement over baselines in some continuous control tasks. However, there are a number of crucial pieces of prior work suggested by reviewers, that I would expect the authors to reference and discuss relation to in the final draft, in particular Bootstrapped DQN Osband et al 2016 which uses disagreement in value space to aid learning in RL.